# Rat Plantar Fascia Stem/Progenitor Cells Showed Lower Expression of Ligament Markers and Higher Pro-Inflammatory Cytokines after Intensive Mechanical Loading or Interleukin-1β Treatment In Vitro

**DOI:** 10.3390/cells12182222

**Published:** 2023-09-06

**Authors:** Wing Sum Siu, Hui Ma, Chun Hay Ko, Hoi Ting Shiu, Wen Cheng, Yuk Wa Lee, Cheuk Hin Kot, Ping Chung Leung, Pauline Po Yee Lui

**Affiliations:** 1Institute of Chinese Medicine, The Chinese University of Hong Kong, Shatin, New Territories, Hong Kong SAR, China; 2State Key Laboratory of Research on Bioactivities and Clinical Applications of Medicinal Plants, The Chinese University of Hong Kong, Shatin, New Territories, Hong Kong SAR, China; 3Department of Orthopaedics and Traumatology, The Chinese University of Hong Kong, Shatin, New Territories, Hong Kong SAR, China; 4Center for Neuromusculoskeletal Restorative Medicine, Hong Kong Science Park, Shatin, New Territories, Hong Kong SAR, China

**Keywords:** plantar fasciitis, plantar fascia stem/progenitor cells, inflammation, intensive mechanical loading

## Abstract

The pathogenesis of plantar fasciitis is unclear, which hampers the development of an effective treatment. The altered fate of plantar fascia stem/progenitor cells (PFSCs) under overuse-induced inflammation might contribute to the pathogenesis. This study aimed to isolate rat PFSCs and compared their stem cell-related properties with bone marrow stromal cells (BMSCs). The effects of inflammation and intensive mechanical loading on PFSCs’ functions were also examined. We showed that plantar fascia-derived cells (PFCs) expressed common MSC surface markers and embryonic stemness markers. They expressed lower *Nanog* but higher *Oct4* and *Sox2*, proliferated faster and formed more colonies compared to BMSCs. Although PFCs showed higher chondrogenic differentiation potential, they showed low osteogenic and adipogenic differentiation potential upon induction compared to BMSCs. The expression of ligament markers was higher in PFCs than in BMSCs. The isolated PFCs were hence PFSCs. Both IL-1β and intensive mechanical loading suppressed the mRNA expression of ligament markers but increased the expression of inflammatory cytokines and matrix-degrading enzymes in PFSCs. In summary, rat PFSCs were successfully isolated. They had poor multi-lineage differentiation potential compared to BMSCs. Inflammation after overuse altered the fate and inflammatory status of PFSCs, which might lead to poor ligament differentiation of PFSCs and extracellular matrix degeneration. Rat PFSCs can be used as an in vitro model for studying the effects of intensive mechanical loading-induced inflammation on matrix degeneration and erroneous stem/progenitor cell differentiation in plantar fasciitis.

## 1. Introduction

Plantar fascia is a thick, ribbon-like fibrous aponeurosis that connects the medial calcaneal tubercle to the heads of the metatarsal bones. Therefore, the plantar fascia is defined anatomically as a ligament. It is divided into three components (central, medial, and lateral). The central component adheres to the underlying flexor digitorum brevis muscle and divides at the level of the mid sole into five bands at the mid-metatarsal level. The plantar fascia contributes to the support of the foot arch by acting as a tie rod, and it undergoes tension when the foot bears weight. The plantar fascia is composed of predominantly longitudinally arranged collagen fibers.

Plantar fasciitis is a degenerative painful musculoskeletal disorder of the plantar fascia that often frustrates patients by disturbing their normal daily activities and reducing their quality of life [1]. It is caused by repetitive trauma at its origin on the calcaneus [2]. In the United States (US), approximately 10% people are predicted to develop heel pain during their lifetime [3] and more than two million individuals are treated annually [4]. The estimated cost for the diagnosis and treatment of plantar fasciitis was approximately USD 284 million in 2007 [5]. Although plantar fasciitis is more prevalent in women aged 40–60 and in physically active individuals such as runners and military personnel [6,7,8], individuals with pes planus and lower medial longitudinal arch [2,7,8,9] are also commonly affected. There is still no standard treatment for plantar fasciitis. It is commonly managed by conservative treatments and occasionally by surgical procedures (open or endoscopic release of plantar fascia) with sub-optimal outcomes [10]. Recurrence of pain is common, and many cases may turn chronic. According to a Cochrane review, there is limited evidence upon which to base the treatment of plantar fasciitis due to the lack of understanding of the underlying pathogenesis [3]. Basic biological research on the pathogenesis of plantar fasciitis is extremely limited due to the lack of in vitro and in vivo models.

Although the plantar fascia is anatomically defined as a ligament, the injury mechanism and the clinical and histological presentation of plantar fasciitis are similar to tendinopathy. It is exposed to repetitive tensile stress induced by cyclic loads associated with daily activities, such as walking and running. The pathogenic mechanisms therefore are expected to share similarities to tendinopathy, despite the fact that it is anatomically defined as a ligament. Plantar fasciitis is marked by tissue pain, inflammation, matrix degeneration, thickening and fibrosis of the plantar fascia, hypervascularity, increased fibroblasts, chondroid metaplasia, and calcification [11,12]. Similar to tendinopathy, overuse or change of loading produces microscopic tears and chronic inflammation, which results in tissue degeneration, has been suggested as an important pathogenic mechanism in plantar fasciitis [13,14,15]. Plantar fasciitis might be initiated by an inflammatory event that subsides and is then overtaken by the degenerative changes in the plantar fascia [8].

Among different non-exclusive pathogenic mechanisms of tendinopathy, altered fate of stem cells after tendon injury has been suggested as an important pathogenic mechanism [16,17]. The functions of tendon-derived stem/progenitor cells (TDSCs) isolated from tendinopathy patients were altered [18,19,20]. More TDSCs, with a lower proliferative capacity and tenogenic potential, but with a higher cellular senescence, non-tenocyte differentiation potential, and inflammatory response, were present in an inflammatory collagenase-induced degenerative failed tendon healing animal model [21,22] and clinical samples of tendinopathy [18,19,20]. Tendinopathic TDSCs were unable to differentiate into tenocytes following mechanical stretch [18]. We hypothesized that the fate of the resident plantar fascia stem/progenitor cells would be altered after mechanical overuse similar to tendinopathy and contribute to the pathogenesis of plantar fasciitis. A recent study has reported the isolation of stem/progenitor cells from human plantar fascia and that the stem/progenitor cells were sensitive to mechanical loading [23]. Intensive mechanical loading increased the expression of non-ligament markers and decreased the expression of ligament markers in the human plantar fascia stem/progenitor cells in that study [23]. Most patients with plantar fasciitis are managed by conservative treatment and hence it is difficult to collect clinical samples for understanding its pathogenesis and developing effective treatment strategies. The rat is an important animal model for biomedical research. The successful isolation of stem/progenitor cells from rat plantar fascia would facilitate the study of their roles in the pathogenesis and therapeutic treatment of plantar fasciitis in the laboratory setting. There has been no report on the isolation and characterization of stem/progenitor cells from rat plantar fascia. This limits biological research in this area. The effects of intensive mechanical loading on the functions of stem/progenitor cells isolated from rat plantar fascia are unknown.

Previous studies on tendinopathy have shown that the tenogenic properties of tendon stem/progenitor cells were compromised in an inflammatory environment [22,24,25]. Interleukin-1 beta (IL-1β) has been implicated as an important mediator of the tissue inflammatory response of tendinopathy [26,27,28]. Although the pathogenesis of plantar fasciitis remains unclear, overuse-induced chronic inflammation has been suggested as an important pathogenic mechanism [13,14,15]. We speculated that the ligament differentiation potential of the resident plantar fascia stem/progenitor cells would be compromised (as shown by the decreased expression of ligament markers and altered expression of non-ligament markers) under overuse-induced inflammation and hence impaired plantar fascia healing. The effects of IL-1β, which was used to mimic the inflammatory condition during plantar fascia overuse, on the expression of inflammatory cytokines, matrix-remodeling markers, ligament markers, and non-ligament markers of plantar fascia stem/progenitor cells have not been explored.

This study therefore aimed to isolate and characterize rat plantar fascia stem/progenitor cells (PFSCs) with reference to bone marrow stromal cells (BMSCs). The effects of inflammation and intensive mechanical loading on PFSCs’ functions were also examined. We hypothesized that PFSCs can be isolated from rat plantar fascia. We further speculated that the expression of ligament markers in PFSCs decreased, while the expression of pro-inflammatory cytokines and extracellular matrix degrading enzymes in PFSCs increased and the expression of non-ligament markers was altered after IL-1β treatment and intensive mechanical loading. A better understanding of the effects of inflammation and intensive mechanical loading on the functions of PFSCs would shed a light on the pathogenesis and hence accelerate the development of novel effective therapeutics for the treatment of plantar fasciitis.

## 2. Materials and Methods

### 2.1. Animals

Ten 8-week-old male Sprague Dawley rats weighing from 250 to 260 g obtained from the Laboratory Animal Services Centre of the Chinese University of Hong Kong (CUHK) were used in this study. Stem cells were isolated from the bone marrow and plantar fascia of the animals after euthanasia. This study was approved by the Animal Research Ethics Committee of the CUHK (AEEC approval number: 14/094/MIS).

### 2.2. Cell Culture Media

Growth medium was prepared by supplementing Minimum Essential Medium α (αMEM) with 10% fetal bovine serum (FBS) and 1% Penicillin/Streptomycin (PS) (Life Technologies, Hewlett, NY, USA).Control medium for the differentiation assays was prepared by supplementing Dulbecco’s Low Glucose Modified Eagle Medium (LG-DMEM) (Sigma-Aldrich, St. Louis, MO, USA) with 10% FBS and 1% PS.Osteogenic medium was prepared by supplementing LG-DMEM with 1 nM dexamethasone, 50 µM ascorbic acid, and 20 mM β-glycerophosphate disodium salt hydrate (BGP) (all from Sigma-Aldrich, St. Louis, MO, USA).Adipogenic medium was prepared by supplementing LG-DMEM with 500 nM dexamethasone, 0.5 mM isobutylmethylxanthine, 50 µM indomethacin, and 10 µg/mL insulin (all from Sigma-Aldrich, St. Louis, MO, USA).Chondrogenic medium was prepared by supplementing LG-DMEM with 100 nM dexamethasone, 50 µg/mL L-ascorbic acid-2-phosphate, 40 µg/mL proline, and 100 µg/mL pyruvate (all from Sigma-Aldrich, St. Louis, MO, USA), 1:100 diluted ITS + Premix (6.25 mg/mL insulin, 6.25 mg/mL transferrin, 6.25 mg/mL selenious acid, 1.25 mg/mL bovine serum albumin, and 5.35 mg/mL linoleic acid) (Becton Dickinson, Franklin Lakes, NJ, USA), 500 ng/mL bone morphogenetic protein (BMP-2) and 10 ng/mL transforming growth factor beta 3 (TGF-β3) (both from R&D Systems, Minneapolis, MN, USA).

### 2.3. Isolation and Culture of Rat PFSCs and BMSCs

To isolate the nucleated cells from the plantar fascia, the plantar muscle was excised and the plantar fascia was then exposed (Figure 1A). Only the central portion of the plantar fascia was harvested (Figure 1B). The plantar fascia of both limbs was then digested with 0.05% trypsin-EDTA for 5 min, followed by further digestion with 1.5 mg/mL type I collagenase (Sigma-Aldrich, St. Louis, MO, USA) for 3 h (Figure 1C). The digested tissues were then passed through a 70 mm cell strainer (Becton-Dickinson, Franklin Lakes, NJ, USA) to yield a single-cell suspension. All plantar fascia-derived cells were resuspended in working αMEM and cultured in a humidified atmosphere at 37 °C, 5% CO_2_. On day 7, all the non-adherent cells were removed and the cells were cultured until the plate was at half confluence. The monolayer of adherent cells was trypsinized, re-plated at 3.64 cells/cm^2^, 36.4 cells/cm^2^ and 364 cells/cm^2^ (passage 1 (P1)), and cultured in a humidified atmosphere at 37 °C, 5% CO_2_. The culture medium was changed twice a week.

On day 14, the colonies formed were stained with 1% crystal violet (Sigma Aldrich, St. Louis, MO, USA) in methanol for 30 min. The number of colonies was counted. Colonies that were less than 2 mm in diameter and those that were faintly stained were ignored. The optimal initial cell density was chosen based on the following criteria: (1) the colony size was not affected by colony-to-colony contact inhibition, and (2) the greatest number of colonies per plantar fascia-derived nucleated cell was obtained. The colonies isolated from the two plantar fasciae of each rat at the optimal initial plating density and the colonies from two rats were pooled as one batch. The cells were sub-cultured with a change of medium every 2–3 days for the experiments. The adherent cells derived from the plantar fascia were named as plantar fascia-derived cells (PFCs) before they were fully characterized as plantar fascia stem/progenitor cells (PFSCs) using the colony-forming assay, immunophenotyping of mesenchymal stromal cell (MSC) surface markers, and multi-lineage differentiation assay in this manuscript. The name “PFSCs” was hence used only for reporting the results of IL-1β stimulation and intensive mechanical loading.

BMSCs were isolated according to our established procedures [29,30]. Briefly, the bone marrow tissues were isolated from the medullary canal of the femora by centrifugation. The mononuclear cells were resuspended in growth medium, plated at the density of 2 × 10^2^/100 mm petri dish (3.64 cells/cm^2^) for the isolation of stem cells, and cultured in a humidified atmosphere at 37 °C, 5% CO_2_ to form colonies. On day 7, all the non-adherent cells were removed and the cells were cultured.

PFCs/PFSCs and BMSCs at P3–P5 were used in all the assays in this study.

### 2.4. Colony-Forming Assay

The clonogenicity of PFCs and BMSCs was compared. Briefly, 2 × 10^2^ PFCs and BMSCs at P3 were seeded in 100 mm sterile petri dishes and cultured for 14 days. The colonies formed were stained and counted as mentioned above. There were 8 samples in each group.

### 2.5. Cell Counting

The proliferative potential of PFCs and BMSCs was assessed by cell counting and immunocytochemical staining of Ki67 described below.

Cells at P3 were plated at 5 × 10^4^ cells in 75 cm^2^ flasks and incubated in a humidified atmosphere at 37 °C, 5% CO_2_. On day 2, 4, 6, and 8, the number of cells in each flask was counted after trypan blue exclusion of dead cells. There were 3 samples at each time point in each group.

### 2.6. Analysis of Immunophenotypes

The immunophenotypes of PFCs and BMSCs were characterized as previously described [30]. A total of 1 × 10^4^/cm^2^ PFCs or BMSCs at P3-P5 were cultured in 150 cm^2^ flasks until confluence. The cells were then trypsinized, resuspended, and incubated with the PE-, FITC-, or APC-conjugated primary antibodies against CD29, CD90, CD31, CD45, and CD71 at 1:50 dilution for 1 h at 4 °C. Isotype controls were included in the study. After washing with PBS, the stained cells were resuspended in 500 µL of ice-cold PBS supplemented with 1% sodium azide and analyzed using the flow cytometer (CANTO II, BD Biosciences, San Jose, CA, USA). The fluorescent signal was analyzed using the FACs Diva software (Version 6.1.3, BD Biosciences, San Jose, CA, USA). A gating strategy based on Forward Scatter (FS) vs. Side Scatter (SS) was employed. A total of 10,000 cells were counted using the flow cytometer. The following primary antibodies were obtained from Bio-Rad, Hercules, CA, USA: hamster anti-mouse/rat PE-conjugated CD29 (Cat.: MCA2298PE), mouse anti-rat PE-conjugated CD90 (Cat. No.: MCA47PE), mouse anti-rat FITC-conjugated CD31 (Cat. No.: MCA1334F), and mouse anti-rat PE-conjugated CD71 (Cat. No.: MCA155PE). Mouse anti-rat APC-conjugated CD45 (Cat. No.: 17-0461-82) was obtained from Invitrogen, Waltham, MA, USA. There were 3 samples for each marker.

### 2.7. Immocytochemical Staining of Stemness Markers and Proliferation Marker

Immunocytochemical staining of the isolated cells was performed. Briefly, 2 × 10^4^ cells at P3 were seeded in the wells of the Nunc^®^ Lab-Tek^®^ Chamber Slide™ system (Sigma-Aldrich, St. Louis, MO, USA) overnight at 37 °C and 5% CO_2_. The cells were then washed with PBS, fixed in 4% paraformaldehyde for 15 min, permeabilized with 0.25% Triton X-100 for 10 min, and blocked with 5% normal goat serum for 1 h. Afterwards, the cells were incubated with mouse anti-rat Ki67 (ab279653), rabbit anti-rat Nucleostemin (NS) (ab70346) (both from Abcam, Cambridge, UK), rabbit anti-rat Nanog (NBP1-77109), Oct4 (NB100-2379SS) or Sox2 (NB110-37235SS) (all from Novus Biologicals, Centennial, CO, USA) (1:200 all) overnight at 4 °C. Primary antibody was replaced with blocking solution in the negative controls. After washing with PBS, the cells were incubated with goat anti-mouse IgG conjugated with Alexa Fluor 488 (A-11001, Invitrogen; for Ki67), goat anti-rabbit IgG conjugated with Alexa Fluor 488 (ab150077, Abcam; for Oct4, Nanog and Sox2) or conjugated with Alexa Fluor 594 (ab150080, Abcam; for NS) (1:100 all) for 30 min at room temperature. After washing with PBS, the cells were stained and mounted with Aqueous Mounting Medium with DAPI (sc-24941, Santa Cruz Biotechnology, Dallas, TX, USA). The stained cells were examined under the fluorescence microscope equipped with a UV laser (IX71, Olympus, Tokyo, Japan). Images were captured with a digital microscope camera (DS-Fi3, Nikon, Tokyo, Japan). There were 3 samples in each group.

### 2.8. mRNA Expression of Stemness Markers

The mRNA expression of stemness markers including *Nanog*, *Oct4* and *Sox2* in PFCs and BMSCs at P3 was compared using quantitative real-time reverse transcription-polymerase chain reaction (qRT-PCR) according to our well-established procedures [31]. Briefly, the target mRNA was amplified in a reaction mixture containing One Step Quanti Fast SYBR Green RT-PCR reaction cocktail (Qiagen, Santa Clarita, CA, USA) and specific primers for *Nanog*, *Oct4*, *Sox2* or *Gapdh* (Table 1, all from Life Technologies, Hewlett, NY, USA) using the CFX96 Touch Real-Time PCR Detection system (Bio-Rad, Hercules, CA, USA). The gene expression was normalized by the expression of the *Gapdh* gene. We have compared the stability of *Gapdh* and *Actb*. Both housekeeping genes showed stable expression in our experiment. We finally chose to normalize the data with *Gapdh*. There were 6 samples in each group.

### 2.9. Multi-Lineage Differentiation Potential

To assess their tri-lineage differentiation potential, PFCs and BMSCs at P3-P5 were seeded at a density of 1 × 10^5^ cells/well in 12-well plates. After culturing the cells for 3 days in standard culture medium, the cells were switched to differentiation media, including osteogenic, adipogenic, chondrogenic, or control medium (Gibco StemPro™ differentiation kits, all from Life Technologies, Hewlett, NY, USA). The differentiation media were changed twice a week, and after 21 days, the cells were fixed using methanol for 10 min and stained with different dyes for 10 min. Specifically, Alizarin Red S (2% in miliQ water) was used to stain osteocytes, Oil Red O (0.3% in isopropanol) was used to stain differentiated adipocytes, and Alcian blue (3% in 3% glacial acetic acid) was used to stain chondrocytes. These dyes bind to calcium deposits, cytoplasmic lipids, and extracellular glycosaminoglycans, respectively. The cells were then washed with PBS and examined under a light microscope. Following image acquisition, the dyes used for staining were extracted using different solvents. Specifically, 10% cetylpyridinium chloride (CPC) was used to extract Alizarin Red S [32], 100% isopropanol was used to extract Oil Red O, and DMSO was used to extract Alcian blue. The color intensity of Alizarin red S, Oil red-O, and Alcian blue was measured at optical density 540, 492, and 450 nm, respectively. In addition, the mRNA expression of osteogenic markers (*Runx2*, *Bglap*, and *Spp1*), adipogenic markers (*Pparg*, *Cfd*, and *Fabp4*), and chondrogenic markers (*Col2a1* and *Sox9*) was quantified using qRT-PCR. The primer sequences are shown in Table 1. There were at least 6 samples in each group.

### 2.10. Expression of Ligament Markers

PFCs and BMSCs at P3 were seeded in 12-well plates at a density of 1 × 10^5^ cells/well. The mRNA expression of *Col1a1*, *Col3a1*, *Eln*, *Scx*, *Tnc*, and *Tnmd* (Table 1) in PFCs and BMSCs was assessed by qRT-PCR, as described above. Previous studies have used these genes as ligament markers [33,34,35]. There were at least 6 samples in each group.

### 2.11. Effects of IL-1β

Inflammation was induced in PFSCs and BMSCs by IL-1β treatment. IL-1β (Life Technologies, Hewlett, NY, USA) at 10 ng/mL was added to PFSCs and BMSCs for 48 h. The concentration of IL-1β used to induce inflammation in PFSCs was selected based on the literature on MSCs [24,36,37]. We have tested the effects of IL1-β on PFSCs at 24 h and 48 h, and 48 h gave a better result for the gene markers tested. The mRNA expression of ligament markers (*Col1a1*, *Col3a1*, *Eln*, *Scx*, *Tnc*, and *Tnmd*), osteogenic markers (*Bglap*, *Runx2*, and *Spp1*), chondrogenic markers (*Col2a1* and *Sox9*), adipogenic markers (*Pparg*, *Cfd*, and *Fabp4*), inflammatory cytokines (*Tnfa*, *Cox2*, *Il6*, *Il10*, and *Il33*), and matrix-remodeling markers (*Mmp1*, *Mmp3*, and *Timp1*) was assessed by qRT-PCR, as described above. The fold change in gene expression normalized to the expression of the housekeeping gene, *Gapdh*, and the mean expression of the untreated control group was calculated using the 2^−ΔΔCT^ formula. The primer sequences are shown in Table 1. There were 6 samples in each group.

### 2.12. Effects of Intensive Mechanical Loading

Cyclic uniaxial stretching was used to induce overuse injury to PFSCs using the in vitro mechanical loading system (ST-140-10, B-Bridge International, Inc., Tokyo, Japan) according to our established protocol [38]. The cells were seeded in specially designed silicone culture plates coated with gelatin at 1 × 10^4^/cm^2^. After confluence, the cells were washed with PBS and stretched at 0.5 Hz at 4% or 8% for 5 h in serum-free medium. Cells without stretching (0%) served as controls. Immediately after stretching, cells were harvested for the assessment of mRNA expression of pro-inflammatory cytokines, matrix remodeling markers, ligament markers, osteogenic markers, chondrogenic markers, and adiogenic markers using qRT-PCR (Table 1). The fold change in gene expression normalized to the expression of the housekeeping gene, *Gapdh*, and the mean expression of the unloaded control group was calculated using the 2^−ΔΔCT^ formula. There were 3–6 samples in each group.

### 2.13. Data Analysis

The colonies isolated from the two plantar fasciae of each rat at the optimal initial plating density and the colonies from two rats were pooled as one batch. The number of samples in each experiment were the technical replicates of the cells from one batch. We conducted and repeated the experiments with different batches of cells. Flow cytometry data were shown in histogram. Cell count data at each time point were presented as mean ± SD and were shown as line graph. qRT-PCR data were presented in box plots or bar charts. The difference between two independent groups were tested by unpaired *t*-test. The effects of intensive mechanical loading on the mRNA expression of different markers were tested by ANOVA followed by Bonferroni post hoc comparison if the data were normally distributed as shown by the Shapiro–Wilk test. Otherwise, the intensive mechanical loading data were tested by Kruskal–Wallis test followed by Dunn’s test. All statistical analyses were performed with GraphPad Prism 6.0 (GraphPad Software, Version 6.0, La Jolla, CA, USA). *p* < 0.05 was considered as statistically significant.

## 3. Results

### 3.1. Optimal Seeding Density for PFSC Isolation

The adherent plantar fascia-derived cells (PFCs) formed the highest numbers of colonies at the seeding density of 2 × 10^2^/100 mm petri dish (3.64 cells/cm^2^) (Figure 2A,B). The cell morphology of PFCs was similar to BMSCs (Figure 2C).

### 3.2. Clonogenicity

The clonogenicity of PFCs and BMSCs at P3 was compared. There were significantly more colonies in the PFC culture compared to the BMSC culture (46.25 ± 27.13 versus 12.88 ± 8.5, *p* < 0.01) (Figure 3A).

### 3.3. Proliferative Potential

PFCs and BMSCs were cultured in 75 cm^2^ flasks at an initial seeding density of 5 × 10^4^ cells/flask and the cell number was counted every 2 days for 8 days. Our results demonstrated that PFCs proliferated significantly faster than BMSCs from day 4 to day 8 (*p* < 0.05, <0.001, <0.001 on day 4, 6, and 8, respectively) (Figure 3B). Nearly all PFCs and BMSCs expressed Ki67 (Figure 3C).

### 3.4. Immunophenotypes

PFCs were positive for the MSC surface markers including CD29 (99.7 ± 0.1%) and CD90 (99.6 ± 0.05%) (Figure 4A). However, they were negative for the endothelial cell marker CD31 (0.0%), hematopoietic stem cell marker CD45 (0.0%), and erythroid lineage marker CD71 (0.0%) (Figure 4A). Similar results were observed for BMSCs (100.0 ± 0.1% for CD29; 99.9 ± 0.05% for CD90; 0.0% for CD31; 0.2 ± 0.01% for CD45; 0.5 ± 0.01% for CD71) (Figure 4B).

### 3.5. Protein and mRNA Expression of Stemness Markers

Immunocytochemical staining revealed that stemness markers, including Nanog, Oct4, Sox2, and NS, were expressed in both PFCs and BMSCs (Figure 5A). We compared the mRNA expression of *Nanog*, *Oct4*, and *Sox2* between PFCs and BMSCs using qRT-PCR. Our results showed that the expression of *Nanog* in PFCs was significantly lower than that in BMSCs (*p* < 0.01) (Figure 5B). However, the expression of *Oct4* and *Sox2* was significantly higher in PFCs compared to that in BMSCs (*p* < 0.01) (Figure 5B).

### 3.6. Multi-Lineage Differentiation Potential

#### 3.6.1. Osteogenic Differentiation

Fewer calcium nodules were formed in PFCs compared to BMSCs after osteogenic induction for 21 days in vitro as indicated by Alizarin red S (Figure 6(Ai)). The quantification of the extracted Alizarin red S dye showed that the OD of PFCs was 3.3-fold lower than that of BMSCs (*p* < 0.001) (Figure 6(Aii)). qRT-PCR analysis also revealed that the mRNA expression of *Bglap*, *Runx2*, and *Spp1* in PFCs was, respectively, 2.8-, 4-, and 1.5-fold lower than that in BMSCs (all *p* < 0.001) (Figure 6(Aiii)).

#### 3.6.2. Adipogenic Differentiation

Fewer oil droplets were formed in PFCs compared to that in BMSCs after adipogenic induction for 21 days in vitro as indicated by Oil Red-O staining (Figure 6(Bi)). Quantification of the extracted Oil Red-O dye showed that the OD of PFCs was 1.6-fold lower than that of the BMSCs (*p* < 0.001) (Figure 6(Bii)). This observation was supported by a 3.05-, 3.94-, and 2.43-fold lower mRNA expression of *Pparg*, *Cfd*, and *Fabp4*, respectively, in PFCs compared to that in BMSCs (all *p* < 0.001) (Figure 6(Biii)).

#### 3.6.3. Chondrogenic Differentiation

After chondrogenic induction for 21 days, there was a higher production of proteoglycans in the PFC group and BMSC group as shown by Alcian blue staining (Figure 6(Ci)). The quantification of the extracted Alcian Blue dye showed that the OD of PFCs was 2.6-fold higher than that of BMSCs (*p* < 0.001) (Figure 6(Cii)). The mRNA expression of chondrogenic markers including *Col2a1* and *Sox9* was significantly higher in PFCs compared to that in BMSCs, namely, 4.2-fold (*p* < 0.001) and 1.32-fold (*p* < 0.01) higher, respectively (Figure 6(Ciii)).

#### 3.6.4. Expression of Ligament Markers

The mRNA expression of ligament markers in PFCs and BMSCs was compared. PFCs showed a significantly higher mRNA expression of *Col1a1*, *Eln*, and *Tnc* (all *p* < 0.01), as well as *Col3a1* and *Tnmd* (both *p* < 0.001). However, there was no significant difference in the mRNA expression of *Scx* between PFCs and BMSCs (Figure 7).

### 3.7. Effects of IL-1β on the Expression of Ligament, Inflammatory, Matrix-Remodeling, and Non-Ligament Markers

IL-1β was used to mimic the inflammatory condition during plantar fascia overuse. Our results showed that IL-1β significantly increased the mRNA expression of the pro-inflammatory cytokines (*Tnfa*, *Cox2*, and *Il6*) but downregulated or had no effect on the expression of anti-inflammatory cytokines (*Il10* (*p* < 0.05) and *Il33* (*p* > 0.05), respectively) in PFSCs (Figure 8A). IL-1β significantly upregulated the expression of *Mmp3* and *Timp1* (both *p* < 0.001) but downregulated the expression of *Mmp1* without statistical significance (*p* = 0.112) in PFSCs (Figure 8B). Although the *Mmp1*/*Timp1* ratio in the IL-1β-treated group was about 0.4-fold of the untreated group, the *Mmp3*/*Timp1* ratio increased by more than 25-fold after IL-1β treatment (Figure 8B). All ligament markers were significantly downregulated in PFSCs after treatment with IL-1β (Figure 8C). The expression of osteogenic (Figure 8D) and chondrogenic (Figure 8E) markers in PFSCs was reduced after IL-1β treatment, with a significant difference for *Runx2* (*p* < 0.001), *Spp1* (*p* < 0.01), and *Col2a1* (*p* < 0.001), while there was no difference in the expression of adipogenic markers (Figure 8F).

### 3.8. Effects of Intensive Mechanical Loading on PFSCs

Intensive mechanical loading of PFSCs at 4% or 8% increased the mRNA expression of all the pro-inflammatory cytokines tested (post hoc *p* < 0.05 for *Il1b* and *Nos2*; post hoc *p* < 0.01 for *Tnfa*, *Il6*, and *Mcp1*; post hoc *p* < 0.001 for *Cox2*) (Figure 9A). The mRNA expression of *Mmp1* (post-hoc *p* < 0.01) and *Mmp3* (post-hoc *p* < 0.05) increased but there was no significant difference in the expression of *Timp1* after intensive mechanical loading (*p* > 0.05) (Figure 9B). The *Mmp3*/*Timp1* ratio also significantly increased after intensive mechanical loading (post hoc *p* < 0.01) (Figure 9B). Except *Col3a1*, *Scx*, and *Tnc*, which showed no significant differences among groups, the expression of ligament markers decreased after intensive mechanical loading (post hoc *p* < 0.05 for 0% vs. 4% loading for *Col1a1* and *Mkx*; post hoc *p* < 0.01 for 0% vs. 8% loading for *Col1a1* and *Mkx*; post hoc *p* < 0.001 for *Eln*, *Tnmd*, and *Dcn*) (Figure 9C). The expression of *Runx2* (0% vs. 4% loading: post hoc *p* < 0.05; 0% vs. 8% loading: post hoc *p* < 0.01) and *Spp1* (0% vs. 4% loading: post hoc *p* < 0.05) decreased after intensive mechanical loading (Figure 9D). The mRNA expression of *Sox9* (post hoc *p* < 0.001) and *Acan* (post hoc *p* < 0.05) increased at 8% loading compared to the unloaded control (Figure 9E). Except *Fabp4*, the mRNA expression of all adipogenic markers tested decreased significantly after intensive mechanical loading (post hoc *p* < 0.01 for 0% vs. 8% loading for *Pparg*; post hoc *p* < 0.01 for 0% vs. 4% loading for *Cebpa*; post hoc *p* < 0.001 for the other markers) (Figure 9F).

## 4. Discussion

We have successfully isolated stem/progenitor cells from plantar fasciae of rats. Similar to BMSCs, PFSCs were adherent to plastic, formed colonies, expressed the common MSC surface markers and some stemness markers, as well as undergoing osteogenic, adipogenic, and chondrogenic differentiation upon induction in vitro. PFSCs proliferated faster and formed more colonies compared with BMSCs. However, they had poor osteogenic and adipogenic differentiation potential upon induction compared to BMSCs. Intensive mechanical loading and IL-1β increased the expression of inflammatory cytokines aincludeincludetrix-degrading enzymes, and reduced the expression of ligament markers in PFSCs.

The expression of stemness markers including Nanog, Sox2, Oct4, and NS was examined in PFCs/PFSCs and BMSCs. Previous studies have shown that MSCs expressed these stemness markers [39,40,41,42] in addition to The International Society for Cellular Therapy’s minimal set of criteria for defining an MSC [43]. Nanog is a transcription factor expressed by inner cell mass, primitive germ cells, and embryonic stem cells (ESCs). It helps to maintain the undifferentiated state of ESCs and promote cell proliferation [44]. Oct4 is a transcription factor, which binds to the DNA of octamer modes. Sox2 transactivates or inhibits the promoters of various target genes to regulate various physiological processes [45]. The ectopic expression of Oct4 and Sox2, together with the other two Yamanaka factors Klf4 and c-Myc, transforms mouse embryonic fibroblasts into induced pluripotent stem cells (iPSCs) [46]. NS, located in the cellular nucleolus, is a guanosine triphosphate binding protein. It regulates the proliferation, differentiation, cell cycle process, and renewal of stem cells and tumor cells [47]. The forced overexpression of NS in cultured cardiac stem cells significantly increased their cell proliferation [48]. It is often used as a stem/progenitor cell marker [42,48,49]. Our results showed that PFCs/PFSCs expressed these stemness markers in addition to meeting the ISCT criteria for defining MSCs.

Although stem cells originating from different tissues shared some common stem cell characteristics, they also exhibited some tissue-specific properties and hence functional differences [50]. Our results showed that PFSCs showed a higher yield, a colony-forming ability, a proliferation and chondrogenic differentiation potential, as well as a higher expression of ligament markers compared to BMSCs. However, they showed a lower osteogenic and adipogenic differentiation potential and hence lower plasticity compared to BMSCs. The plasticity of a stem cell is associated with its regenerative potential [51,52,53,54]. For instance, pluripotent stem cells derived from multipotent MSCs were reported to stay in the body longer to modulate immune responses and have a greater potential to improve muscle recovery in rotator cuff tears after transplantation compared to multipotent adult MSCs [51]. The reprogramming of unipotent terminally differentiated cells into multipotent stem cells increased their tissue regenerative potential [52]. Mandible MSCs showed a higher tri-lineage differentiation potential compared to femur MSCs, and the result was correlated with their enhanced in vivo bone regenerative capacity after transplantation [53]. Moreover, the loss of multi-potency in aged stem cells also contributed to ageing and age-related disease [54]. In addition, tendon stem/progenitor cell clones derived from single cells that could differentiate into four lineages (tendon, cartilage, bone, and adipose) showed a higher expression of scleraxis and mohawk as well as gel contraction upon in vitro tenogenesis, compared to clonal cell lines that could only differentiate into tri-lineages (tendon, cartilage, and bone, or tendon, cartilage, and adipose), suggesting that clonal cell lines with a higher multi-lineage differentiation potential might have improved therapeutic benefits for tendon repair [55]. We therefore speculate that the poor plasticity of PFSCs compared to BMSCs tested in this study might contribute to the poor regenerative capacity of plantar fascia after injury. Further studies are needed to confirm our hypothesis.

Plantar fasciitis is common in physically active individuals and there is still no standard treatment at present. As tissue-specific stem/progenitor cells residing in the plantar fascia, PFSCs are expected to play important roles in maintaining tissue homeostasis and repair of the injured plantar fascia. A better understanding of the functions of PFSCs might shed a light on the physiology of the plantar fascia as well as the pathogenesis and effective treatments of plantar fasciitis. Although the pathogenesis of plantar fasciitis remains unclear, overuse-induced microscopic tears and chronic inflammation, resulting in tissue degeneration, which has been suggested as an important pathogenic mechanism [13,14,15]. We hypothesized that the ligament differentiation potential of the resident plantar fascia stem/progenitor cells might be compromised (as shown by the decreased expression of ligament markers and altered expression of non-ligament markers) similarly to tendinopathy after mechanical overuse. The accumulation of micro-injuries in repeated trauma due to the poor ligament regenerative capacity of PFSCs, particularly under an inflammatory condition after overuse, might contribute to the development of failed healing and hence plantar fasciitis. To examine our hypotheses, we therefore investigated the effects of IL-1β and intensive mechanical loading on the gene expression of inflammatory cytokines, matrix-remodeling markers, ligament markers, and non-ligament markers of PFSCs in this study. Our results showed that intensive mechanical loading reduced the expression of ligament markers, osteogenic markers, and adipogenic markers, but increased the expression of chondrogenic markers, inflammatory cytokines, and matrix-degrading enzymes in rat PFSCs, indicating an altered fate and increased inflammation of PFSCs after intensive exercise. Our results were different from the results of a previous study, which reported that human PFSCs expressed higher levels of non-ligament markers (*RUNX2*, *LPL*, and *COL2A1*) after high-intensity mechanical loading for 12 h [23]. Only one osteogenic, one adipogenic, and one chondrogenic marker were measured in the previous study. The loading duration was different in the present study (5 h), which might have contributed to the differences. However, the altered fate of PFSCs with a reduced expression of ligament markers was demonstrated in both studies, supporting the negative impact of overuse on PFSC properties. The reduced expression of adipogenic markers and increased expression of chondrogenic markers in PFSCs after intensive mechanical loading were different from our findings after IL-1β treatment. This might be due to the presence of different inflammatory environment in the two conditions.

Inflammation is a key component in the pathogenesis of plantar fasciitis [8,11,14,56]. Our results showed that intensive mechanical loading increased the expression of *IL1b* and IL-1β increased the expression of pro-inflammatory cytokines (*Tnfa*, *Cox2*, and *Il6*) and matrix-remodeling markers (*Mmp3* and *Timp1*) in PFSCs. Although the mRNA expression of *Timp1* also increased after IL-1β treatment, which might be an attempt of the cells to suppress excessive matrix degradation mediated by *Mmp3*, the ratio of *Mmp3*/*Timp1* increased more than 28.4-fold after IL-1β treatment. In addition, IL-1β reduced the expression of ligament markers in PFSCs in this study. Intensive mechanical loading increased the expression of all pro-inflammatory cytokines tested (*Mmp1*, *Mmp3*, and *Mmp3*/*Timp1* ratio) in our rat PFSCs in this study, which is consistent with the results reported in human PFSCs that intensive mechanical loading increased the protein expression of Cox-2 and production of IL-6 and PGE2, as well as the mRNA expression of *MMP1* and *MMP2* [23]. Hence, the production of inflammatory cytokines after overuse or degenerative injury of the plantar fascia might suppress the ligament differentiation of PFSCs, resulting in failed healing and pain.

Most patients with plantar fasciitis undergo conservative treatment and it is difficult to obtain clinical samples for understanding the pathogenesis and developing novel treatments. The availability of a valid animal model and an in vitro cell culture model would facilitate advancements in the field. The successful isolation of PFSCs and the results showing the reduced expression of ligament markers under inflammation and after high-intensity mechanical loading suggested that PFSCs could be used as an in vitro model for studying the pathogenesis and treatment of plantar fasciitis.

There are some weaknesses in this study. First, we only tested the effects of overuse after intensive mechanical loading at 4% and 8% for 5 h. The effects of other loading regimens on PFSCs should be examined in the future. Second, we only examined the mRNA expression of inflammatory cytokines, ligament, non-ligament, and matrix-remodeling makers after IL-1β or intensive mechanical loading. Further studies should examine these markers at the protein level. Third, due to the small size of the rat plantar fascia, we did not separate the core and sheath parts for PFSC isolation. A previous study has reported that stem/progenitor cells in the human plantar fascia sheath and core showed different stemness, proliferation rate, and expression of extracellular matrix proteins and endothelial markers [23]. Therefore, our rat PFSCs are expected to exhibit properties of both stem/progenitor cells of plantar fascia sheath and core, and the study of structural proteins and angiogenesis of plantar fascia using rat PFSCs should be cautioned. Finally, the injury pattern and mechanisms of rat and human plantar fasciae may be different. Hence, rat PFSCs cannot totally replace human PFSCs in the study of plantar fasciitis. Since most patients with plantar fasciitis undergo conservative treatment, it is difficult to obtain clinical samples for the study of the disease pathogenesis and treatment. Our results showed that both intensive loading and inflammation reduced the expression of ligament markers and increased the expression of inflammatory cytokines in rat PFSCs, which was similar to the findings in the stem/progenitor cells isolated from the human plantar fascia sheath and core [23]. This suggested that rat PFSCs are a valid in vitro model for studying the effects of intensive mechanical loading-induced inflammation on matrix degeneration and erroneous stem/progenitor cell differentiation in plantar fasciitis.

## 5. Conclusions

In conclusion, rat PFSCs were successfully isolated. They proliferated faster and formed more colonies compared to BMSCs. However, they had a poor multi-lineage differentiation potential compared to BMSCs. Inflammation after overuse altered the fate and inflammatory status of PFSCs, which might lead to a poor ligament differentiation of PFSCs and extracellular matrix degeneration. Rat PFSCs can be used as an in vitro model for studying the effects of intensive mechanical loading-induced inflammation on matrix degeneration and erroneous stem/progenitor cell differentiation in plantar fasciitis.

## Figures and Tables

**Figure 1 cells-12-02222-f001:**
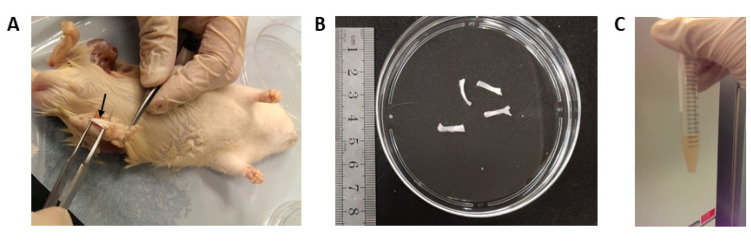
Procedures for the isolation of stem cells from the plantar fascia of rats. Photographs showing the (**A**) exposure of the plantar fascia; (**B**) dissection of the middle portion of the plantar fascia; and (**C**) digestion of the isolated tissue with type I collagenase. Arrow: plantar fascia.

**Figure 2 cells-12-02222-f002:**
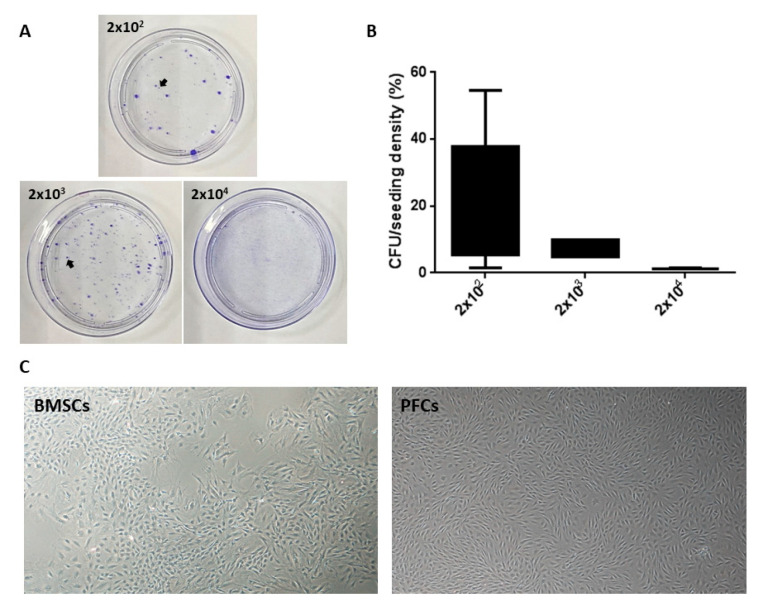
Optimal initial seeding density for the isolation of plantar fascia stem/progenitor cells (PFSCs) and cell morphology of isolated cells. (**A**) Photomicrographs and (**B**) boxplot showing the clonogenicity of adherent plantar fascia-derived cells (PFCs) at P3 stained with 1% crystal violet at different initial seeding densities (3.64 cells/cm^2^, 36.4 cells/^2^, and 364 cells/cm^2^). Arrows indicated the positive colonies that were 2 mm in diameter and sharply stained. PFCs formed the highest numbers of colonies at 3.64 cells/cm^2^ (2 × 10^2^ in 100 mm dish). This cell density was used in all subsequent experiments. (**C**) Photomicrographs showing the morphology of BMSCs and PFCs at P3. Magnification: 100×; scale bar: 50 µm.

**Figure 3 cells-12-02222-f003:**
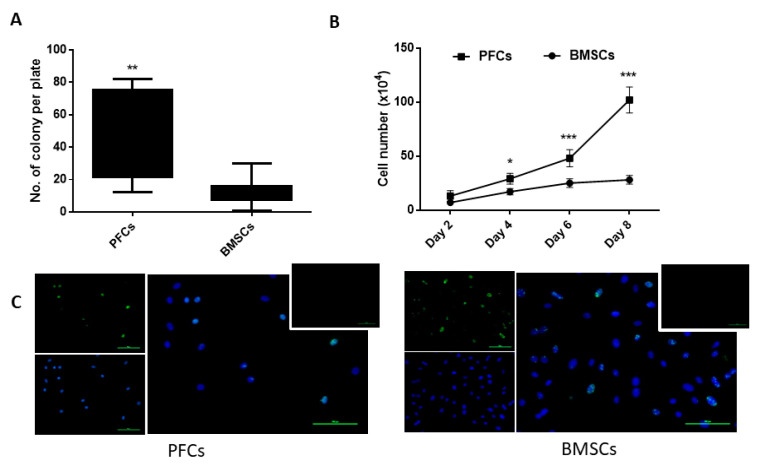
Clonogenicity and proliferative potential of PFCs compared to BMSCs. (**A**) Boxplot showing the colony-forming ability of PFCs and BMsCs at P3. n = 8/group. (**B**) Graph showing the growth of PFCs compared to BMSCs at different times of culture at P3-P5. n = 3/group; * *p* < 0.05; ** *p* < 0.01; *** *p* < 0.001. (**C**) Immunocytochemical staining of Ki67 in PFCs compared to BMSCs (upper left: Ki67, lower left: DAPI, right: merged, upper right: negative control). Magnification: 200×; scale bar: 100 µm; n = 3/group.

**Figure 4 cells-12-02222-f004:**
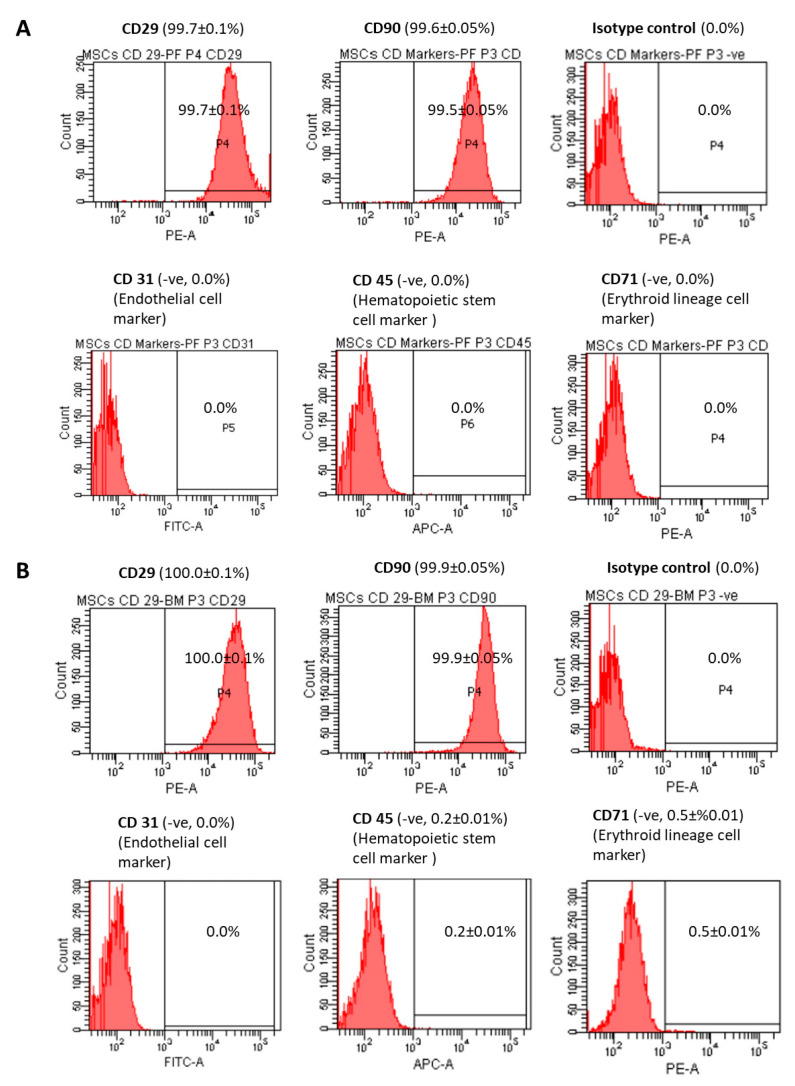
Immunophenotypes of PFCs and BMSCs. Histograms showing the expression of CD29, CD90, CD31 (-ve), CD45 (-ve), CD71 (-ve) and the corresponding isotype control in (**A**) PFCs and (**B**) BMSCs. The percentage of cells showing positive expression is shown in brackets. n = 3/group.

**Figure 5 cells-12-02222-f005:**
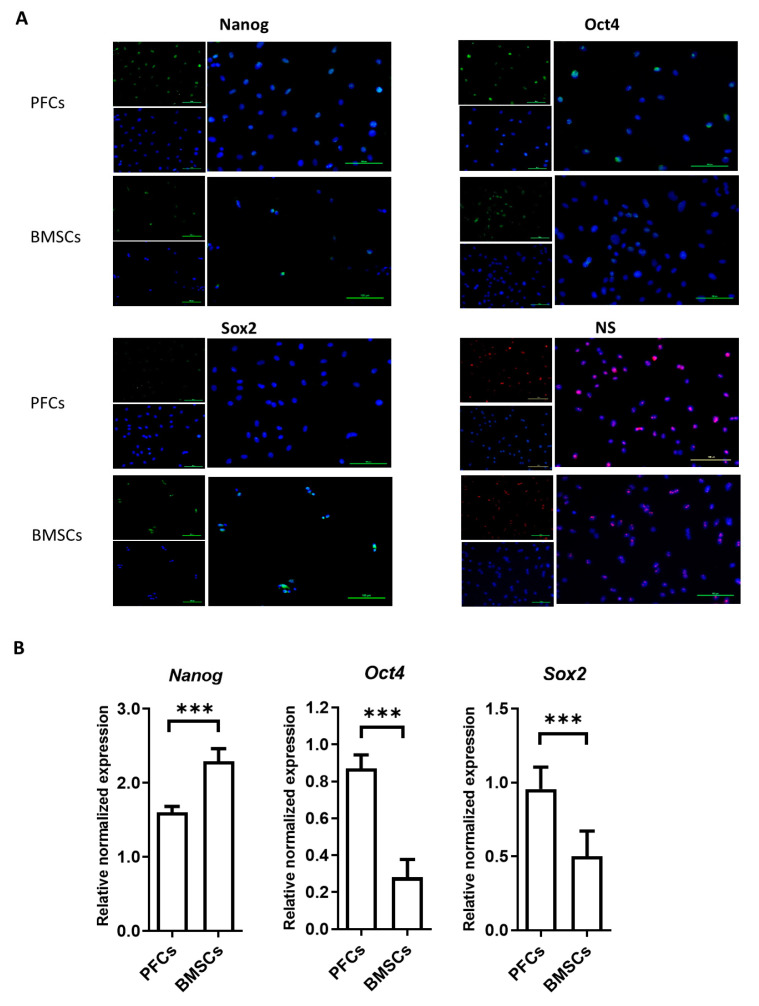
Expression of stemness markers in PFCs compared to BMSCs. (**A**) Photomicrographs showing the immunocytochemical staining of Nanog, Oct4, Sox2, and nucleostemin (NS) (upper left: primary antibody, lower left: DAPI, right: merged). Magnification: 200×; scale bar: 100 µm; n = 3/group. (**B**) Bar charts showing the mRNA expression of *Nanog*, *Oct4*, and *Sox2* in PFCs and BMSCs. *** *p* < 0.001; n = 6/group.

**Figure 6 cells-12-02222-f006:**
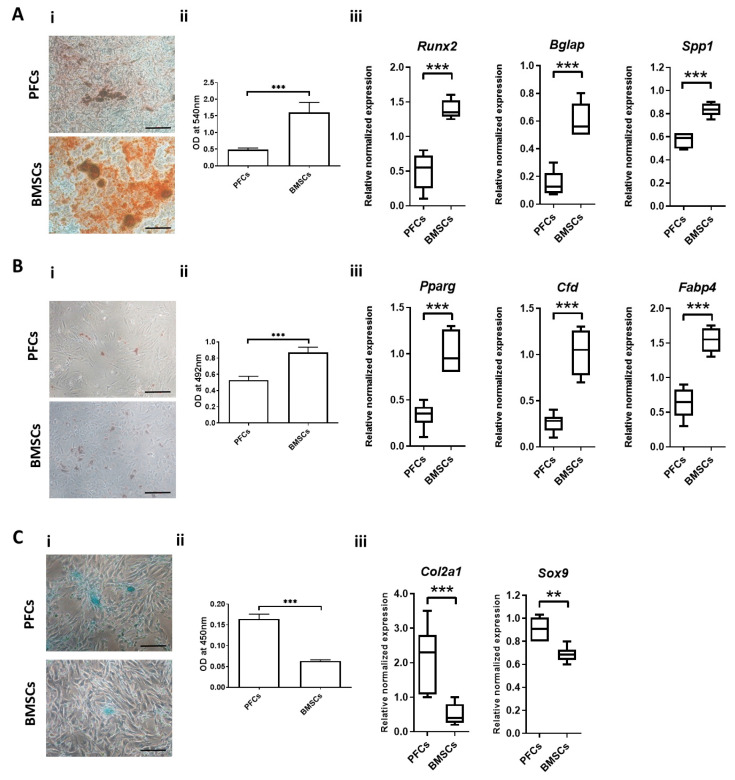
Multi-lineage differentiation potential of PFCs compared to BMSCs. (**A**) Osteogenic differentiation of PFCs and BMSCs after induction for 21 days in vitro. (**i**) Photomicrographs showing calcium nodule formation as indicated by Alizarin red S staining after induction for 21 days. Magnification: 100×; scale bar: 50 µm. (**ii**) OD of Alizarin red S from PFCs and BMSCs. (**iii**) Boxplots showing the mRNA expression of osteogenic markers on day 7. (**B**) Adipogenic differentiation of PFCs and BMSCs after induction for 21 days in vitro. (**i**) Photomicrographs showing oil droplet formation as indicated by Oil red-O staining after induction for 21 days. Magnification: 100×; scale bar: 50 µm. (**ii**) OD of Oil red-O from PFCs and BMSCs. (**iii**) Boxplots showing the mRNA expression of adipogenic markers on day 7. (**C**) Chondrogenic differentiation of PSCs and BMSCs after induction for 21 days in vitro. (**i**) Photomicrographs showing proteoglycan formation as indicated by Alcian Blue staining after induction for 21 days. Magnification: 200×; scale bar: 25 µm. (**ii**) OD of Alcian Blue from PFCs and BMSCs. (**iii**) Boxplots showing the mRNA expression of chondrogenic markers in PFCs and BMSCs after chondrogenic induction for 21 days in vitro. ** *p* < 0.01; *** *p* < 0.001; n ≥ 6/group.

**Figure 7 cells-12-02222-f007:**
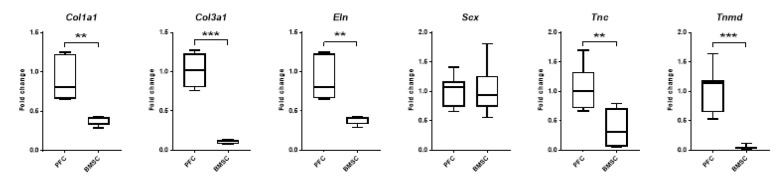
Expression of ligament markers in PFCs compared to BMSCs. Boxplots showing the mRNA expression of ligament markers between PFCs and BMSCs. ** *p* < 0.01; *** *p* < 0.001; n ≥ 6/group. The *x*-axis represents the cell types. The *y*-axis represents the fold change of mRNA expression normalized to the expression of the reference gene, *Gapdh*, and the mean expression of the PFC group.

**Figure 8 cells-12-02222-f008:**
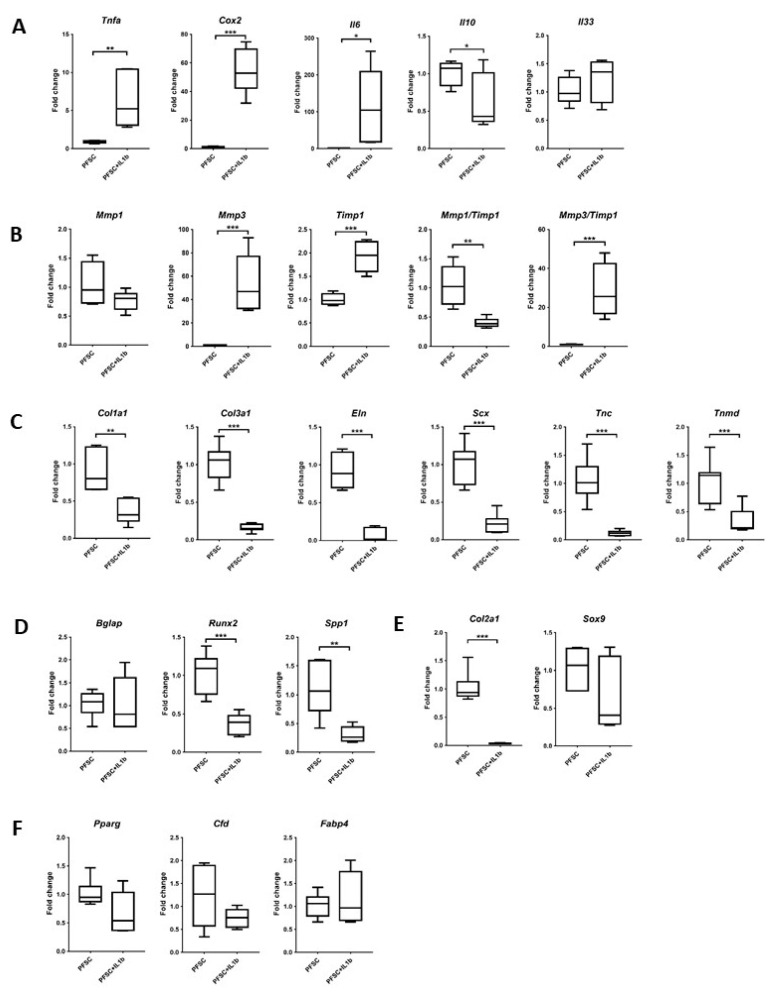
Expression of inflammatory, matrix-remodeling, ligament, and non-ligament markers in PFSCs after treatment with IL-1β. Boxplots showing the mRNA expression of (**A**) pro- and anti-inflammatory markers; (**B**) matrix-degrading enzymes and their inhibitor; (**C**) ligament markers; (**D**) osteogenic markers; (**E**) chondrogenic markers; and (**F**) adipogenic markers. * *p* < 0.05; ** *p* < 0.01; *** *p* < 0.001; n ≥ 6/group. The *x*-axis represents the treatment groups. The *y*-axis represents the fold change of mRNA expression normalized to the expression of the reference gene, *Gapdh*, and the mean expression of the untreated control group.

**Figure 9 cells-12-02222-f009:**
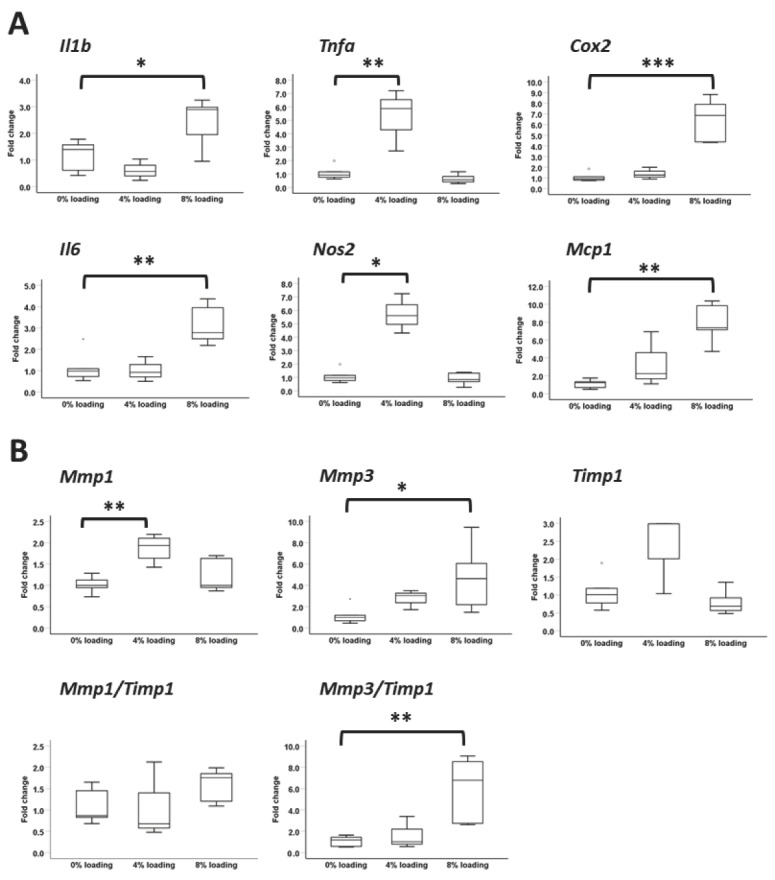
Expression of inflammatory, matrix-remodeling, ligament, and non-ligament markers in PFSCs after intensive mechanical loading. Boxplots showing the mRNA expression of (**A**) princludenflammatory markers; (**B**) matrix-degrading enzymes and their inhibitor; (**C**) ligament markers; (**D**) osteogenic markers; (**E**) chondrogenic markers; and (**F**) adipogenic markers. * *p* < 0.05; ** *p* < 0.01; *** *p* < 0.001; n = 3–6/group; “o” and “*” above and below each box of the boxplot are outliner and extreme value, respectively. The x-axis represents the treatment groups. The y-axis represents the fold change of mRNA expression normalized to the expression of the reference gene, *Gapdh*, and the mean expression of the unloaded control group. *Nos2*, *Mcp1*, *Timp1*, *Mmp3*/*Timp1*, *Mkx*, *Spp1*, and *Cfd* were analyzed by non-parametric Kruskal–Wallis test while the rest of the markers were analyzed by ANOVA.

**Table 1 cells-12-02222-t001:** Primer sequences for qRT-PCR.

Gene	Primer Nucleotide Sequence	Accession Number
*Nanog*	(F) GCCACCCACACTTGTGACTA (R) TTCTCGCCTGTGTGAGTTCG	NM_053713
*Oct4*	(F) GTCCCTAGGTGAGTCGTCCT (R) TGGAAGCTTAGCCAGGTTCG	NM_001009178
*Sox2*	(F) GAGGAGGAGAGCGACTGTTT (R) CTGGCGGAGAATAGTTGGGG	NM_001109181
*Runx2*	Primer 1: (F) CACAAGTGCGGTGCAAACTT (R) GCAGCCTTAAATATTACTGCATGG Primer 2: (F) CCGATGGGACCGTGGTT (R) CAGCAGAGGCATTTCGTAGCT	NM_053470 NM_053470.1
*Spp1*	(F) CCGAGGTGATAGCTTGGCTT (R) CTCTTCATGCGGGAGGTGAG	NM_012881
*Bglap*	(F) ATGAGGACCCTCTCTCTGCT (R) AGGTAGCGCCGGAGTCTATT	NM_013414
*Pparg*	Primer 1: (F) CCTGTTGACCCAGAGCATGG (R) GGTCCACAGAGCTGATTCCG Primer 2: (F) CGGCGATCTTGACAGGAAAG (R) GCTTCCACGGATCGAAACTG	NM_013124 AB019561
*Cfd*	(F) TGGGGCAATCACCAAGAACA (R) CGAGATCCCCACGTAACCAC	NM_001077642
*Fabp4*	(F) TCGTCATCCGGTCAGAGAGT (R) CCAGCTTGTCACCATCTCGT	U75581.1
*Col2a1*	(F) GTTCACGTACACTGCCCTGA (R) AAGGCGTGAGGTCTTCTGTG	NM_012929
*Sox9*	Primer 1: (F) TGGGAGCGACAACTTTACCA (R) GAGGAGGAGGGAGGGAAAAC Primer 2: (F) AGAGCGTTGCTCGGAACTGT (R) TCCTGGACCGAAACTGGTAAA	XM_001081628 XM_343981.2
*Col1a1*	Primer 1: (F) CCCAGCGGTGGTTATGACTT (R) GGGTTTGGGCTGATGTACCA Primer 2: (F) CATCGGTGGTACTAAC (R) CTGGATCATATTGCACA	NM_053304.1 NM_053356.1
*Col3a1*	Primer 1: (F) TTCCTGGGAGAAATGGCGAC (R) ACCAGCTGGGCCTTTGATAC Primer 2: (F) TGCAATGTGGGACCTGGTTT (R) GGGCAGTCTAGTGGCTCATC	NM_032085 NM_032085.1
*Eln*	Primer 1: (F) GGAAAGTTCCTGGTGTCGGT (R) TCCAGCACCATACTTCGCTG Primer 2: (F) GCTTAGGAGTCTCAACAGGTGC (R) CGGAACCTTGGCCTTGACTC	NM_012722 NM_012722.1
*Scx*	Primer 1: (F) GCCTGTGGGGACCTAAAGAG (R) AGCATGAACACGACAGGGTT Primer 2: (F) AACACGGCCTTCACTGCGCTG (R) CAGTAGCACGTTGCCCAGGTG	NM_001130508 NM_001130508.1
*Tnc*	Primer 1: (F) CCACAGAAGCTGAACCGGAA (R) CAGTATCCGTCCCATCCACG Primer 2: (F) AAAGCAGCCACCCGCTATTAC (R) GGATCTCCTCTGTCAAGACCTCAA	NM_053861 NM_053861.2
*Tnmd*	Primer 1: (F) CACCTCAGCAGTGGTCTCTC (R) TGTGCTCCATGCCATAGGTC Primer 2: (F) GTGGTCCCACAAGTGAAGGT (R) GTCTTCCTCGCTTGCTTGTC	NM_022290 NM_022290.1
*Tnfa*	Primer 1: (F) CAGCCGATTTGCCATTTCATAC (R) GGCTCTGAGGAGTAGACGATAA Primer 2: (F) AAATGGGCTCCCTCTCATCAGTTC (R) TCTGCTTGGTGGTTTGCTACGAC	NM_012675.3
*Cox2*	Primer 1: (F) TCTCCAACCTCTCCTACTACAC (R) CTCCACCGATGACCTGATATTT Primer 2: (F) TGTATGCTACCATCTGGCTTCGG (R) GTTTGGAACAGTCGCTCGTCATC	NM_011198.4 S67722.1
*Il6*	Primer 1: (F) ATCTGCCCTTCAGGAACAGC (R) AGCCTCCGACTTGTGAAGTG Primer 2: (F) TCCTACCCCAACTTCCAATGCTC (R) TTGGATGGTCTTGGTCCTTAGCC	NM_012589.2
*Il10*	(F) TCCGGGGTGACAATAACTGC (R) GCAGCTGTATCCAGAGGGTC	NM_012854.2
*Il33*	(F) GACCAGCTATCTCCCATCACT (R) TTGGATACTGCCAAGCAGGG	NM_001014166.1
*Mmp1*	(F) CCACTAACATTCGAAAGGGTTT (R) GGTCCATCAAATGGGTTATTG	NM_001134530.1
*Mmp3*	(F) ATGGGCCTGGAATGGTCTTG (R) CCCTCCATGAAAAGACTCAGAGG	NM_133523.3
*Timp1*	(F) CAGCAAAGGCCTTCGTAAA (R) TGGCTGAACAGGGAAACACT	NM_053819.1
*Il1b*	(F) CACCTCTCAAGCAGAGCACAG (R) GGGTTCCATGGTGAAGTCAAC	NM_031512.2
*Nos2*	(F) TCCTCAGGCTTGGGTCTTGTTAG (R) TTCAGGTCACCTTGGTAGGATTTG	NM_012611.3
*Mcp1*	(F) GGCCTGTTGTTCACAGTTGCT (R) TCTCACTTGGTTCTGGTCCAGT	M57441.1
*Mkx*	(F) TTTACAAGCACCGTGACAACCC (R) ACAGTGTTCTTCAGCCGTCGTC	XM_017600733.2
*Dcn*	(F) GTTCTGATCTGGGTCTGGACAAAG (R) CTTAAAGGCCCCCTCTTTGATC	NM_024129.1
*Acan*	(F) CTTGGGCAGAAGAAAGATCG (R) GTGCTTGTAGGTGTTGGGGT	NM_022190.1
*Cebpa*	(F) AAGGCCAAGAAGTCGGTGGA (R) CAGTTCGCGGCTCAGCTGTT	NM_012524.2
*Lpl*	(F) GTACAGTCTTGGAGCCCATGC (R) CCAAGCCAGTAATTCTATTGACCTTC	NM_012598.2
*Gapdh*	(F) CTCAGTTGCTGAGGAGTCCC (R) ATTCGAGAGAAGGGAGGGCT	NM_017008.4

## Data Availability

All data generated or analyzed during this study are included in this published article.

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
