# Peer review of "Rat Plantar Fascia Stem/Progenitor Cells Showed Lower Expression of Ligament Markers and Higher Pro-Inflammatory Cytokines after Intensive Mechanical Loading or Interleukin-1β Treatment In Vitro"

_cells, 2023, doi:10.3390/cells12182222_

Round 1

Reviewer 1 Report (Previous Reviewer 1)

The authors have undertaken a thorough revision

Author Response

Reply: Thank you very much for your comments on our manuscript.

Reviewer 2 Report (New Reviewer)

This study is very interesting, but can cells derived from rat plantar fasciitis be used to study the pathogenesis of human plantar fasciitis?

Previous studies, ref. 23 revealed two types of cells in human PF that have different structures such as collagenous tissue and blood vessels and exhibit different biological properties such as clonogenicity, pluripotency and proliferative potential.

Are rat PFCs or PFSCs more similar to human PF-S or PF-C cells?

An explanation of the disadvantages or limitations of using rat PFCs to study plantar fasciitis of tissues composed of PF-S and PF-C in humans should be added.

Author Response

This study is very interesting, but can cells derived from rat plantar fasciitis be used to study the pathogenesis of human plantar fasciitis?

Reply: Rat is a commonly used animal model for the study of pathogenesis and treatment of tendinopathy in various anatomical sites [1]. Our team and other researchers have shown that tissue resident tendon-derived stem /progenitor cells (TDSCs) play a crucial role in the pathogenesis of tendinopathy. The function of TDSCs isolated from patients were altered [2-4]. More TDSCs, but with a lower proliferative capacity and tenogenic potential, and a higher cellular senescence, non-tenocyte differentiation potential and inflammatory response, were presented in clinical samples of tendinopathy [2-4]. Similar results were observed in TDSCs isolated from the animal models of tendinopathy [5-6]. Therefore, TDSCs isolated from animals are useful for studying the pathogenesis and novel interventions of tendinopathy.

Since the injury mechanism, clinical and histological presentation of plantar fasciitis are similar to tendinopathy, its pathogenic mechanisms therefore are expected to share similarities to tendinopathy. In this study, we demonstrated that both inflammation and in vitro tensile mechanical loading (mimicking ligament overuse) altered the fate and inflammatory status of PFSCs, and the cellular response was similar to the data reported in human plantar fascia stem cells reported in the literature [7], supporting that rat PFSCs can be used a model for elucidating the underlying mechanisms and potential treatment of human plantar fasciitis.

References:

  1. Lui PP, Maffulli N, Rolf C, Smith RK. What are the validated animal models for tendinopathy? Scand J Med Sci Sports. 2011 Feb;21(1):3-17.
  2. Chang W, Callan KT, Dragoo JL. The Behavior of Tendon Progenitor Cells from Tendinopathic Tendons: Implications for Treatment. Tissue Eng Part A. 2020 Jan;26(1-2):38-46.
  3. Still C 2nd, Chang WT, Sherman SL, Sochacki KR, Dragoo JL, Qi LS. Single-cell transcriptomic profiling reveals distinct mechanical responses between normal and diseased tendon progenitor cells. Cell Rep Med. 2021 Jul 21;2(7):100343.
  4. Dakin SG, Buckley CD, Al-Mossawi MH, Hedley R, Martinez FO, Wheway K, Watkins B, Carr AJ. Persistent stromal fibroblast activation is present in chronic tendinopathy. Arthritis Res Ther. 2017 Jan 25;19(1):16.
  5. Rui YF, Lui PP, Wong YM, Tan Q, Chan KM. Altered fate of tendon-derived stem cells isolated from a failed tendon-healing animal model of tendinopathy. Stem Cells Dev. 2013 Apr 1;22(7):1076-85.
  6. Hu JJ, Yin Z, Shen WL, Xie YB, Zhu T, Lu P, Cai YZ, Kong MJ, Heng BC, Zhou YT, Chen WS, Chen X, Ouyang HW. Pharmacological Regulation of In Situ Tissue Stem Cells Differentiation for Soft Tissue Calcification Treatment. Stem Cells. 2016 Apr;34(4):1083-96.
  1. Zhang J, Nie D. Rocha JL, Hogan MV, Wang JH. Characterization of the structure, cells, and cellular mechanobiological response of human plantar fascia. J. Tissue Eng. 2018, 9, 2041731418801103.

Previous studies, ref. 23 revealed two types of cells in human PF that have different structures such as collagenous tissue and blood vessels and exhibit different biological properties such as clonogenicity, pluripotency and proliferative potential.

Are rat PFCs or PFSCs more similar to human PF-S or PF-C cells?

Reply: Our rat PFCs or PFSCs were isolated from the central portion of plantar fasciitis. Due to the small size of rat patellar tendon, we could not separate the core and sheath parts of the plantar fasciitis. Therefore, we expected our rat PFCs or PFSCs to exhibit properties of both PF-S and PF-C cells. While the extent of response and stemness of PF-S and PF-C in the published study were different, both of them were able to form colonies, expressed pluripotency markers and undergo tri-lineage differentiation. Although moderate-intensity loading increased the expression of collagen type I in PF-C but collagen type IV in PF-S and high-intensity loading increased the expression of CD105 in PF-S only, both PF-S and PF-C increased the mRNA expression of MMP-1, MMP-2, LPL, Runx2 and Collagen type II under moderate and high intensity mechanical loading and they both increased the production of Cox-2, PGE2 and IL-6 in response to high-intensity loading. We therefore believe that our rat cells were similar to human plantar fascia stem / progenitor cells in general.

An explanation of the disadvantages or limitations of using rat PFCs to study plantar fasciitis of tissues composed of PF-S and PF-C in humans should be added.

Reply: we believe that our rat PFCs were an advantage, rather than a limitation, for the study of pathogenesis and treatment of plantar fasciitis. Our results showed that rat PFSCs exhibited mesenchymal stem cell properties. They were sensitive to high-intensity mechanical loading and inflammation, with reduced mRNA expression of ligament markers but increased expression of inflammatory cytokines and matrix-degrading enzymes. The results were similar to the response of human of PF-S and PF-C to high-intensity mechanical loading as reported in the literature. Our results supported the hypothesis that inflammation after overuse might alter the fate and inflammatory status, leading to poor ligament differentiation of PFSCs and extracellular matrix degeneration in plantar fasciitis.

Most patients with plantar fasciitis are managed by conservative treatment and hence it is difficult to collect clinical samples for understanding its pathogenesis and developing effective treatment strategies. This explained why there is only one pre-clinical study of plantar fasciitis in the literature in the last 20-30 years. This limits the biological research in this area. Our results showed that rat PFSCs can be used as an in vitro model for studying human plantar fasciitis.

This is discussed in line 569-575 of the manuscript and is quoted below for the reviewer’s information-

“Most patients with plantar fasciitis undergo conservative treatment and it is difficult to get clinical samples for understanding the pathogenesis and developing novel treatments. The availability of a valid animal model and in vitro cell culture model would facilitate the advancement in the field. The successful isolation of PFSCs and the results showing the reduced expression of ligament markers under inflammation and after high intensity mechanical loading suggested that PFSCs could be used as an in vitro model for studying the pathogenesis and treatment of plantar fasciitis.”

This manuscript is a resubmission of an earlier submission. The following is a list of the peer review reports and author responses from that submission.

Round 1

Reviewer 1 Report

This paper attempts to derive and characterise stem cells from the plantar fascia. Although a cell population is isolated, I disagree with the authors with their conclusion that there analysis demonstrates that a stem cell population has been isolated. I also have considerable concerns with the immunocyochemical and histological staining results provided and their interpretation on their data with regard to lineage markers. These concerns are detailed along with others below.

Abstract: lines 28-30 why would poor osteogenic and adipogenic differentiation potential lead to poor healing response?

Graphical abstract – check visibility of “Rat Plantar Fascia”. It is not clear what the blue box with arrows represents?

Methods:

- 20 mM of Beta glycerophosphate is very high for your osteogenic differentiation. Levels above 5 mM commonly induce dystrophic mineralisation – how do you know you were not just seeing this?

Iine 134 – what were the different densities?

Figure 1A) can you add an arrow to highlight the plantar fascia

1B) please add a scale bar.

Lines141-145. Once you had optimised the density to get the maximum number of colonies – how were these colonies passaged? As a bulk culture (i.e. colonies and single cells) or were clonal line established etc?

Section 2.6 – please include the gating strategy and report the flow cytometry methods are per the MIFLOwCyt guidelines https://biotech.ufl.edu/wp-content/uploads/2021/03/nihms76346-1.pdf

Lines 183-184. How long was fixation, permeablisation and blocking performed for?

Lines 206-207. Did you compare GAPDH to other housekeeping genes for stability?

Section 3.9 why did you measure marker gene expression after 7 days but calcium/oil droplets after 21 days?

For all methods section – please clarify if when you say three samples you mean three biological replicates (i.e cells derived from different rats).

Section 3.11 – what was the reason behind the concentration and timing of application of the IL1B?

Section 3.13. Please explain if normality testing of your data was performed and why a Kruskall Wallace test was performed rather than an anova. A Mann-Whitney U test is not suitable as a post hoc test. A Dunn’s test should be used.

Results – it is not clear from figure 2, what you are classing as colonies versus what is faintly stained. Please clarify on on the higher density plates examples of what you included and excluded in your counts.

Lines 265-266. The ability to form a colony does not mean that you have derived stem cells. Please amend this text.

Figure 3 B – please clarify what passage the cells were seeded at for this data.

Figure 3C – this staining is very faint and not very convincing. You should be seeing nuclear staining and your staining is very weak and appears to be cytoplasmic. Please replace with better images and  show the negative controls alongside. You state that nearly all cells expressed Ki67 but it doesn’t look like this for the PFSCs.

Figure 4. These must be the graphs from one replicate. But in the legend you say n=3/group. Please provide a summary of the average % positive cells +/- SD for each marker across the 3 replicates.

Figure 5. These immunocytochemical images do not show correct staining. Nanog, SOX2 and Oct4 are all nuclear proteins and you show cytoplasmic staining. You would need to run an appropriate positive control alongside (e.g. rat iPSCs) to compare to. You also need to include the appropriate negative control images. What is the relevance of nucleostemin? This is not mentioned in the text?

Figure 5 B. It is not clear how these levels of expression compare to a truly pluripotent cell type. Please include iPSCs etc in the analysis.

Section 3.6 – you say fewer nodules and fewer oil droplets were produced but there is no quantitative data shown.

Figure 6 B. The images of the oil droplets are not clear. It looks like only debris has been stained. Higher magnification images showing clear droplet formation should be shown.

Figure 6C. The alcian blue staining is not convincing. You have very few cells in the PFSCs and very little staining in either image. Alcian blue staining is not mentioned in the methods.

Section 3.6.4. What is the justification of choosing these genes are ligament markers? Many of them are commonly used as tendon markers.

Figure 7. What is the x-axis on this graph? Fold change relative to what?

Figure 8. What is the x-axis? If these are undifferentiated cells – why are they expressing markers of osteoblasts, cartilage and adipogenic cells? If they express all these genes – what is the relevance of expressing the “ligament” markers you used previously? What is the relevance of these markers of other lineages going down after IL1B? This same query would also apply to your data in figure 8.

Discussion – opening line. I would disagree that these cells can be characterised as stem cells based on the data presented. You have derived a proliferating cell type but it expresses a wide range of genes. The immunocytochemistry is not convincing and neither is the histology for the bone, fat and cartilage differentiation. So it is not clear what criteria you are using to suggest they are stem cells as opposed to adult cells or progenitor cells?

Other points raised regarding the relevance of other lineage markers and their decrease in response to IL1B should be discussed. The discussion is fairly limited in identifying limitations with the study.

Conclusion: why would a lack of differentiation into other tissues contribute to poor repair?

Reviewer 2 Report

Well written interesting paper on changes in expression of specific genes in in vitro model of plantar fasciitis. Several issues need to be addressed. 

Introduction: it is not clear why plantar fascia is considered to be a ligament rather than being closer to be a tendon – it is an aponeurosis, a fibrous tissue connecting bone with muscle (this makes it more like a tendon). The ligament is noted for higher content of elastin than tendon – have the authors look at the content of elastin and elastic fibers?  A couple of paragraphs later they compare plantar fasciitis to tendinopathy, but later throughout the manuscript they refer to ligament and its problems. If they decide to go with the ligament analogy they need to support with evidence.

Methods: expression of ligament markers: what makes this panel different from a panel for tendons, except for elastin (which does not figure much in the Results anyway)?

Effects of mechanical loading: why were not BMSCs included in mechanical loading, if they are considered a good alternative to use of PFSCs in healing (see in Discussion).  Why were not plantar fasciae used in mechanical loading?

Results: Optimal seeding: Confusing claims. The authors claim that density of 2x102 cells/dish is optimal to get the most colonies, however, I am not convinced by Fig. 2A. The dish with 2x103cells appears to have more, though somewhat smaller colonies, some of them become confluent. Fig. C: the dish with BMSCs cells shows larger cells than the PFSC dish, and quite a few cells which could be cells in mitosis or detached.

Clonogenicity: Fig. 3C: I am not sure what counts as a colony – is it just the light blue spots in the merged photo, or the darker blue spots (likely not), very few colonies in either photo. This is not convincing. The same problem appears in Expression of pluripotence markers (Fig. 5) – please explain and/or find better fields on cell cultures.

Multilineage differentiation: Fig. 6B: shows definitely more oil droplets in PFSCs than in BMSCs in contrast to what the authors claim.

Effects of IL-1β…, effects of loading (last two sections of Results): why were not BMSCs included in this section? 

Discussion: In view of necessary revisions and corrections pointed out above, appropriate changes will have to be made in the first two paragraphs of Discussion.